# Tensorized Multi-View Multi-Label Classification via Laplace Tensor Rank

**Qiyu Zhong** [1]   **Yi Shan** [1 2]   **Haobo Wang** [3]   **Zhen Yang** [1]   **Gengyu Lyu** [1]

## Abstract

In multi-view multi-label classification (MVML), each object has multiple heterogeneous views and is annotated with multiple labels. The key to deal with such problem lies in how to capture cross-view consistent correlations while excavate multi-label semantic relationships. Existing MVML methods usually employ two independent components to address them separately, and ignores their potential interaction relationships. To address this issue, we propose a novel Tensorized MVML method named TMvML, which formulates an MVML tensor classifier to excavate comprehensive cross-view feature correlations while characterize complete multi-label semantic relationships. Specifically, we first reconstruct the MVML mapping matrices as an MVML tensor classifier. Then, we rotate the tensor classifier and introduce a low-rank tensor constraint to ensure view-level feature consistency and label-level semantic co-occurrence simultaneously. To better characterize the low-rank tensor structure, we design a new Laplace Tensor Rank (LTR), which serves as a tighter surrogate of tensor rank to capture high-order fiber correlations within the tensor space. By conducting the above operations, our method can easily address the two key challenges in MVML via a concise LTR tensor classifier and achieve the extraction of both cross-view consistent correlations and multi-label semantic relationships simultaneously. Extensive experiments demonstrate that TMvML significantly outperforms state-of-the-art methods.

[1]College of Computer Science, Beijing University of Technology, China [2]Idealism Beijing Technology Co., Ltd., Beijing, China [3]School of Software Technology, Zhejiang University, Ningbo, China. Correspondence to: Gengyu Lyu <lyugengyu@gmail.com>.

*Proceedings of the $42^{nd}$ International Conference on Machine Learning*, Vancouver, Canada. PMLR 267, 2025. Copyright 2025 by the author(s).

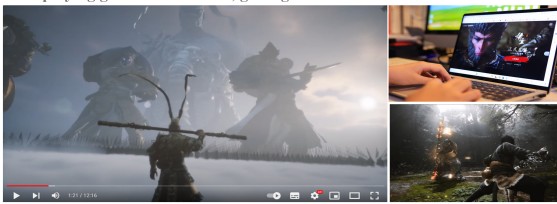

*Figure 1.* An example of MVML webpage classification.

## 1. Introduction

Multi-View Multi-Label Learning (MVML) aims to learn from training data, where each sample is represented by several heterogeneous features while associated with multiple semantic labels (Lyu et al., 2024). Different from single-view multi-label data, MVML data incorporates complementary information from various views, thereby providing more comprehensive descriptions for objects. Such characteristic has naturally led to its extensive applications in many complex data analysis tasks (Liu et al., 2024d; Fu et al., 2024). For example, in news webpage classification (Figure 1), a news webpage can be characterized by *text*, *image* and *video* features, and annotated with multiple class labels such as *Wukong*, *Sales* and *Steam*. MVML provides an effective framework to learn a desired multi-label classifier for bridging heterogeneous features with their corresponding labels and make proper predictions for new webpages.

The key to learn from MVML data lies in how to efficiently integrate heterogeneous features while comprehensively characterize all relevant labels. For example, (Zhao et al., 2023) propose an MVML method called LVSL, which seeks cross-view correlations and multi-label relationships by learning the contribution weights of different views and applying label correlation matrix with low-rank constraint. (Zhong et al., 2024) introduce a non-negative matrix factorization based MVML method called GNAM, which learns individual and common information across different views and leverages a dynamic label correlation matrix to enhance recognition performance. Despite these MVML methods have achieved competitive performance improvements, they still face two main challenges: (1) On one hand, these meth-

ods are formulated by matrix theory, which only focus on capturing one- or second-order relationships between each pair of views and fail to explore higher-order correlations across the whole view space. (2) On the other hand, they separate the extraction of cross-view consistent correlations from multi-label semantic characterization, and ignore the potential interactions between feature representation and semantic expression, inevitably leading to suboptimal results.

To address the above issues, we propose a novel Tensorized Multi-View Multi-Label Classification method via Laplace Tensor Rank (TMvML), which is the first attempt to utilize tensorized MVML classifier to achieve the high-order feature correlations extraction and multi-label semantic relationships characterization simultaneously. Specifically, we first reconstruct the multi-view multi-label mapping matrices into a unified MVML tensor classifier to characterize the structure of multi-dimensional data. Then, we rotate the tensor classifier and introduce a low-rank tensor constraint to capture the high-order information embedded within the mapping tensor, which is achieved by comparing each row (label-specific) and each column (view-specific) of the frontal slices along the third dimension (feature-specific). To better characterize the low-rank tensor structure, we design a novel tensor rank approximation named Laplace Tensor Rank (LTR), which preserves larger singular values and discards smaller ones (removed noise information) to obtain an accurate low-rank tensor representation. Extensive experimental results demonstrate that our proposed TMvML is significantly superior to other state-of-the-art methods. The main contributions of our paper are summarized as:

- We propose a novel tensorized MVML classification method named TMvML, which is the first attempt to design a concise low-rank MVML tensor classifier to excavate cross-view feature correlations and characterize multi-label semantic relationships simultaneously.

- To better characterize the low-rank tensor structure, we design a novel tensor rank approximation named Laplace Tensor Rank (LTR), which preserves larger singular values and discards smaller ones to obtain an accurate low-rank tensor representation.

- Extensive comparable results and detailed experimental analysis have shown that our proposed TMvML method can achieve significantly superior performance against other state-of-the-art methods.

## 2. Related Work

### 2.1. Multi-Label Learning (MLL)

MLL focuses on learning from data with multiple labels, and its challenge lies in how to completely characterize multi-label semantic relationship (Liu et al., 2024a; Zhang & Zhang, 2024). Existing studies mainly employ different strategies to explore label correlations for semantic expression enhancement. For example, (Xie & Huang, 2022) propose a partial multi-label learning method named PML-NI, which captures the linear correlations of the multi-label classifier by a trace norm with low-rank properties. (Sun et al., 2022) propose a method called Global-Local Label Correlation (GLC), which introduces a label coefficient matrix to exploit global label structures across multiple subspaces and a label manifold regularizer to capture local label correlations. (Si et al., 2023) propose a high-rank and high-order method called HOMI, which uses self-representation to keep the rank of the label matrix unchanged and indicate the high-order correlations among labels explicitly.

### 2.2. Multi-View Learning (MVL)

MVL handles the data with multiple heterogeneous view features by exploring consensus and complementary information hidden in different views (Jiang et al., 2021; Zhang et al., 2024; Xu et al., 2024; 2023a;b). Existing MVL methods can be roughly divided into two types based on whether high-order correlation among multi-view representations is explored, including non-tensor methods and tensor-based methods. Non-tensor methods employ matrix-level constraints to explore the point-to-point relationship within one view or each pair of views. For example, (Cao et al., 2015) diversify the model structure using the Hilbert-Schmidt Independence Criterion (HSIC) on pairs of affinity matrices. (Jiang et al., 2024) fuse feature projection and similarity graph to simultaneously select features and learn a unified graph. The tensor-based methods are born to exploit the high-order correlation among views. (Xie et al., 2018) extend the low-rank constraint from the matrix level to the tensor level by minimizing the Tensor Nuclear Norm (TNN) to capture the high-order consistency. To further enhance the characterization of low-rank tensors, (Ji & Feng, 2025) introduce a novel tensor rank approximation called Enhanced Tensor Rank, which is more noisy-robust to explore the high-order consistency among different views.

### 2.3. Multi-View Multi-Label Learning (MVML)

MVML can be seen as an integration of both MVL and MLL, which needs to address both MVL and MLL issues simultaneously (Lu et al., 2023; Liu et al., 2024c; Wei et al., 2025). For example, (Tan et al., 2021) introduce an MVML method called ICM2L, which captures shared patterns through a common subspace and view-specific features with individual classifiers and introduces a label correlation matrix to enhance recognition performance. (Lyu et al., 2024) propose a label-driven view-specific fusion method, which directly encodes individual view features to construct a unified multi-label classifier, and captures label correlations using a transformer-based semantic-aware label graph. (Liu

et al., 2024b) propose an attention-induced MVML method, which weights features by the confidence derived from joint attention, and utilizes the weak label correlation matrix and graph attention network to guide classification.

## 3. The Proposed Method

### 3.1. Notations

Formally speaking, we use bold-uppercase $\mathbf{X}$ for matrices, bold-lowercase $\mathbf{x}$ for vectors and calligraphy letter to denote the tensor $\mathcal{X} \in \mathbb{R}^{n_1 \times n_2 \times n_3}$. For a 3-way tensor $\mathcal{X}$, $\mathcal{X}^k$ is used to represent $k$-th frontal slice; $\mathcal{X}(:,i,j)$, $\mathcal{X}(i,:,j)$ and $\mathcal{X}(i,j,:)$ denote the mode-1, -2, -3 fibers; $\mathcal{X}_f = \mathrm{fft}(\mathcal{X},[],3)$ means the Fast Fourier Transformation (FFT) along the third dimension of tensor $\mathcal{X}$. Given an MVML dataset $\mathcal{D} = \{(\mathbf{X}^v, \mathbf{Y}) | 1 \le v \le V\}$, each $\mathbf{X}^v = [\mathbf{x}_1^v, \mathbf{x}_2^v, \ldots, \mathbf{x}_n^v]^\top \in \mathbb{R}^{n \times d_v}$ represents the feature vectors of $n$ instances under $v$-th view and $\mathbf{Y} \in \{0,1\}^{n \times q}$ is the label matrix.

### 3.2. Formulation

In this paper, we propose a tensorized MVML method named TMvML, which formulates a low-rank MVML tensor classifier to simultaneously extract comprehensive cross-view feature correlations while model complete multi-label semantic relationships. To better characterize the low-rank tensor structure, we develop a new Laplace Tensor Rank (LTR), which serves as a more precise surrogate for tensor rank, enabling a comprehensive exploration of multi-view and multi-label information to improve model performance.

#### 3.2.1. MVML TENSOR CLASSIFIER

In the task of traditional multi-view multi-label learning, the label matrix $\mathbf{Y}$ is usually approximated by a linear mapping $\mathbf{W}^v$ from the feature space $\mathbf{X}^v$:

$$\min_{\mathbf{W}^v} \sum_{v=1}^{V} \|\mathbf{Y} - \mathbf{X}^v \mathbf{W}^v\|_F^2 + \sum_{v=1}^{V} \mathcal{R}(\mathbf{W}^v), \quad (1)$$

where $\mathbf{W}^v = [\mathbf{w}_1^v, \mathbf{w}_2^v, \ldots, \mathbf{w}_{d_v}^v]^\top$ is the $v$-th view mapping matrix. Label co-occurrence is also known to be widely presented in multi-label space (Read et al., 2011), leading to the linear dependency of the feature mapping matrix $\mathbf{W}^v$. Therefore, researchers usually denote $\mathcal{R}(\cdot)$ as the low-rank constraint term to capture such intrinsic property.

Unfortunately, although Eq. (1) may have explored the multi-label semantic relationships, it still treats each view independently, ignoring the feature correlations among different views. To address this limitation, not only should we keep the low-rank constraint for each $\mathbf{W}^v$, but also need to further ensure the consensus principle by imposing low-rank across all views. Motivated by the fact that tensor can characterize the low-rank structure of multi-dimensional

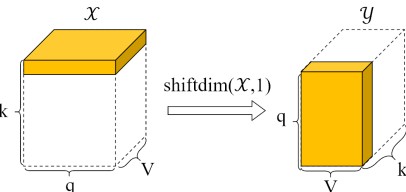

*Figure 2.* The rotated coefficient tensor in our method.

data, we incorporate it into Eq. (1) and obtain the tensorized MVML objective function as:

$$\min_{\mathbf{W}^v, \mathbf{E}, \mathbf{A}^v, \mathbf{Z}^v} \sum_{v=1}^{V} \|\mathbf{Y} - \mathbf{Z}^v \mathbf{W}^v\|_F^2 + \mathcal{T}(\mathcal{W}) + \|\mathbf{E}\|_{2,1}$$
$$s.t. \, \forall v, \, \mathbf{X}^v = \mathbf{Z}^v \mathbf{A}^v + \mathbf{E}^v, \mathbf{A}^v(\mathbf{A}^v)^T = \mathbf{I}, \quad (2)$$
$$\mathcal{W} = \Phi(\mathbf{W}^1, \cdots, \mathbf{W}^V),$$

where the $v$-th view low-dimensional representation matrix $\mathbf{Z}^v \in \mathbb{R}^{n \times k}$ and the basis matrix $\mathbf{A}^v \in \mathbb{R}^{k \times d_v}$ are obtained by matrix factorization. $\mathcal{T}(\cdot)$ is the tensor rank or its approximation. $\mathbf{E}^v \in \mathbb{R}^{n \times d_v}$ denotes reconstruction error. $\Phi(\cdot)$ constructs the tensor classifier $\mathcal{W}$ by merging different classifiers $\mathbf{W}^v \in \mathbb{R}^{k \times q}$ to a 3-mode tensor, and then rotate its dimensionality to $q \times V \times k$, as shown in Figure 2.

**Remark 1. [*The superiority of* $\Phi(\cdot)$]** *Directly applying the tensor low-rank constraint is still not sufficiently effective. To better model low-rank constraints at both the label and view levels within the tensor space, we perform a rotation operation on the coefficient tensor $\mathcal{W} \in \mathbb{R}^{k \times q \times V}$, transforming it into $\mathcal{W} \in \mathbb{R}^{q \times V \times k}$, as illustrated in Figure 2. This rotation repositions the $q \times V$ surface as the frontal slice, enabling the exploration of interactions between different views and labels by comparing every row (label-specific) and every column (view-specific) of the $q \times V$ surface. By constraining all the frontal slices of the tensor, the rotated tensor offers deeper insights, enhancing the exploration of the higher-order correlations of views and labels.*

#### 3.2.2. LAPLACE TENSOR RANK

To better characterize the low-rank tensor structure, we design a novel Laplace Tensor Rank (LTR), defined as follows:

**Definition 1.** Given a tensor $\mathcal{W} \in \mathbb{R}^{n_1 \times n_2 \times n_3}$, then the Laplace Tensor Rank (LTR) is defined as:

$$\|\mathcal{W}\|_{\mathrm{LTR}} = \frac{1}{n_3} \sum_{k=1}^{n_3} \|\mathcal{W}_f^k\|_{\mathrm{LTR}}$$
$$= \frac{1}{n_3} \sum_{k=1}^{n_3} \sum_{i=1}^{h} \left(1 - \exp\left(-\frac{e^\delta \mathcal{S}_f^k(i,i)}{\delta}\right)\right), \quad (3)$$

where $0 < \delta \le 1$, $h = \min(n_1, n_2)$ and $\mathcal{S}_f$ is obtained by t-SVD of $\mathcal{W}_f = \mathcal{U}_f \mathcal{S}_f \mathcal{V}_f^\top$ in Fourier domain.

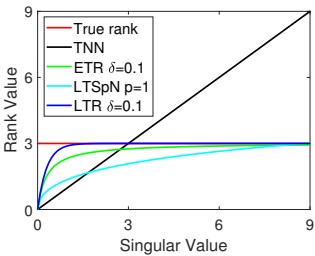

*Figure 3.* The comparisons of different methods to approximate the true rank function as the singular value increases.

**Remark 2. [*The superiority of LTR*]** *The approximation function used by LTR is $f_{LTR}(x) = 1 - \exp\left(-\frac{e^{\delta}x}{\delta}\right)$, which is inspired by the Laplace function (Lu et al., 2015). Basically, $f_{LTR}(0) = 0$ is satisfied, which is consistent with the true rank function. We compare LTR with other existing methods (i.e., TNN (Xie et al., 2018), $LTS_pN$ (Guo et al., 2022) and ETR (Ji & Feng, 2025)). LTR approximates the rank best in Figure 3, especially for larger and near-zero singular values. When $x \rightarrow +\infty$, we obtain $f_{LTR}(x) \rightarrow 1$, which is a better substitute for true rank than TNN, $LTS_pN$ and ETR. If $x \rightarrow 0$, it is easy to prove that $f_{LTR}(x) \approx \frac{e^{\delta^2}x}{\delta+x} \gg \log(1 + x^p) \gg x$, which means LTR achieves a stronger penalization on near-zero singular values. This signifies that $\|\mathcal{W}\|_{LTR}$ can robustly penalize small singular values associated with noise, preserving valuable large singular values and make sure that $\mathcal{W}$ has a spatial low-rank structure to capture high-order fiber correlations.*

### 3.2.3. THE FINAL OBJECTIVE FUNCTION OF TMvML

By integrating Eq. (2) and Eq. (3), our proposed TMvML can be formulated as follows:

$$\min_{\mathcal{W},\mathbf{E},\mathbf{A}^v,\mathbf{Z}^v} \sum_{v=1}^{V} \|\mathbf{S}(\mathbf{Y} - \mathbf{Z}^v\mathbf{W}^v)\|_F^2 + \alpha \|\mathbf{E}\|_{2,1} + \beta \|\mathcal{W}\|_{LTR}$$
$$s.t. \ \forall v, \ \mathbf{X}^v = \mathbf{Z}^v\mathbf{A}^v + \mathbf{E}^v, \mathbf{A}^v(\mathbf{A}^v)^T = \mathbf{I},$$
$$\mathcal{W} = \Phi(\mathbf{W}^1, \cdots, \mathbf{W}^V), \mathbf{E} = [\mathbf{E}^1, \cdots, \mathbf{E}^V], \tag{4}$$

where $\alpha$ and $\beta$ are two trade-off parameters. The vertical concatenation along the column of error matrix, i.e., $\mathbf{E} = \left[\mathbf{E}^1; \mathbf{E}^2; \ldots; \mathbf{E}^V\right]$, can enforce the column of $\mathbf{E}^v$ in each view to have jointly consistent magnitude values (Liu et al., 2012). To ensure the multi-view representation is predictive corresponding to the known labels, we design the filtering matrix $\mathbf{S} \in \mathbb{R}^{n \times n}$ to select the labeled samples with $S_{ii} = 1$ if the $i$-th sample is labeled and $S_{ii} = 0$ otherwise.

### 3.3. Optimization

By utilizing the principles of the alternating direction method of multipliers (ADMM) (Lin et al., 2011), we introduce a separable variable $\mathcal{G}$ to transform Eq. (4) into an unconstrained Lagrangian optimization problem:

$$\mathcal{L}(\{\mathbf{W}^v\}_{v=1}^V, \mathbf{E}^v, \{\mathbf{A}^v\}_{v=1}^V, \mathcal{G}, \{\mathbf{Z}^v\}_{v=1}^V, \mathbf{B}^v, \mathcal{C})$$
$$= \sum_{v=1}^{V} \|\mathbf{S}(\mathbf{Y} - \mathbf{Z}^v\mathbf{W}^v)\|_F^2 + \alpha \|\mathbf{E}\|_{2,1} + \beta \|\mathcal{G}\|_{LTR}$$
$$+ \frac{\rho}{2}\|\mathcal{W} - \mathcal{G}\|_F^2 + \sum_{v=1}^{V}\left(\langle\mathbf{B}^v, \mathbf{X}^v - \mathbf{Z}^v\mathbf{A}^v - \mathbf{E}^v\rangle\right.$$
$$+ \frac{\mu}{2}\|\mathbf{X}^v - \mathbf{Z}^v\mathbf{A}^v - \mathbf{E}^v\|_F^2\Big) + \langle\mathcal{C}, \mathcal{W} - \mathcal{G}\rangle, \tag{5}$$

where tenser $C$ and matrix $\mathbf{B}^v$ are two Lagrange multipliers, $\mu$ and $\rho$ are penalty parameters. Then, we solve the variables in Eq. (5) through the following five subproblems.

**$\mathbf{Z}^v$-subproblem:** We solve the problem by separating the labeled and unlabeled parts, taking advantage of the diagonal property of $\mathbf{S}$. For the labeled part, with other variables fixed and taking the derivative of Eq. (5) with respect to $\mathbf{Z}^v$ to zero, we can obtain:

$$2\mathbf{Y}_l(\mathbf{W}^v)^\top - 2\mathbf{Z}_l^v\mathbf{W}^v(\mathbf{W}^v)^\top + \mathbf{B}_l\mathbf{A}^\top$$
$$+ \mu(\mathbf{X}_l^v\mathbf{A}^\top - \mathbf{Z}_l^v\mathbf{A}\mathbf{A}^\top - \mathbf{E}_l^v\mathbf{A}^\top) = 0, \tag{6}$$

where the subscript $l$ and $u$ indicate variables corresponding to labeled and unlabeled data, respectively. We obtain the updating rule for $\mathbf{Z}_l^v$ as:

$$\mathbf{Z}_l^v = \frac{2\mathbf{Y}_l(\mathbf{W}^v)^\top + \mathbf{B}_l\mathbf{A}^\top + \mu\mathbf{X}_l^v\mathbf{A}^\top - \mu\mathbf{E}_l^v\mathbf{A}^\top}{2\mathbf{W}^v(\mathbf{W}^v)^\top + \mu\mathbf{I}}. \tag{7}$$

Accordingly, for the unlabeled part, we update $\mathbf{Z}_u^v$ by:

$$\mathbf{Z}_u^v = \frac{\mathbf{B}_u\mathbf{A}^\top + \mu\mathbf{X}_u^v\mathbf{A}^\top - \mu\mathbf{E}_u^v\mathbf{A}^\top}{\mu\mathbf{I}}. \tag{8}$$

After obtaining $\mathbf{Z}_l^v$ and $\mathbf{Z}_u^v$, the common representation corresponding to all data $\mathbf{Z}^v$ is obtained as $\mathbf{Z}^v = [\mathbf{Z}_l^v, \mathbf{Z}_u^v]$.

**$\mathbf{E}^v$-subproblem:** With other variables fixed, the subproblem for $\mathbf{E}^v$ can be formulated as

$$\min_{\mathbf{E}} \frac{\alpha}{\mu}\|\mathbf{E}\|_{2,1} + \frac{1}{2}\|\mathbf{E} - \hat{\mathbf{E}}\|_F^2, \tag{9}$$

where $\hat{\mathbf{E}}$ is constructed by horizontally concatenating the matrices $\mathbf{X}^v - \mathbf{Z}^v\mathbf{A}^v + \frac{1}{\mu}\mathbf{B}^v$ together along column. According to (Liu et al., 2019), the solution can be achieved by applying the $\ell_{2,1}$ minimization thresholding operator,

$$\mathbf{E}_{:,i}^* = \begin{cases} \frac{\|\hat{\mathbf{E}}_{:,i}\|_2 - \frac{\alpha}{\mu}}{\|\hat{\mathbf{E}}_{:,i}\|_2}\hat{\mathbf{E}}_{:,i}, & \|\hat{\mathbf{E}}_{:,i}\|_2 > \frac{\alpha}{\mu} \\ 0 & \text{otherwise.} \end{cases} \tag{10}$$

where $\hat{\mathbf{E}}_{:,i}$, represents the $i$-th column of $\hat{\mathbf{E}}$.

**$\mathcal{G}$-subproblem:** With other variables fixed, the subproblem for $\mathcal{G}$ can be formulated as

$$\min_{\mathcal{G}} \beta \|\mathcal{G}\|_{LTR} + \frac{\rho}{2}\left\|\mathcal{G} - (\mathcal{W} + \frac{\mathcal{C}}{\rho})\right\|_F^2. \tag{11}$$

**Theorem 3.1.** *Suppose $\mathcal{A} \in \mathbb{R}^{n_1 \times n_2 \times n_3}$ with t-SVD $\mathcal{A} = \mathcal{U} * \mathcal{S} * \mathcal{V}^T$ and $\gamma > 0$. The Laplace Tensor Rank Minimization problem can be described as follows,*

$$\min_{\mathcal{G}} \gamma \|\mathcal{G}\|_{LTR} + \frac{1}{2}\|\mathcal{G} - \mathcal{A}\|_F^2. \qquad (12)$$

*Then, the optimal solution $\mathcal{G}^*$ is obtained as,*

$$\mathcal{G}^* = \mathcal{U} * ifft(Prox_{f,\gamma}(\mathcal{S}_f), [], 3) * \mathcal{V}^\top, \qquad (13)$$

*where $ifft(Prox_{f,\gamma}(\mathcal{S}_f), [], 3) \in \mathbb{R}^{n_1 \times n_2 \times n_3}$ is an f-diagonal tensor, and $Prox_{f,\gamma}(\mathcal{S}_f^k(i,i))$ satisfies,*

$$\arg\min_{x \geq 0} \frac{1}{2}(x - \mathcal{S}_f^k(i,i))^2 + \gamma f(x), \qquad (14)$$

*where $f(x) = 1 - \exp\left(-\frac{e^\delta x}{\delta}\right)$.*

*Proof.* In the Fourier domain, according to the linearity of FFT and the fact that $\|\mathcal{G}\|_F^2 = \frac{1}{n_3}\|\mathcal{G}_f\|_F^2$, the objective function $\frac{1}{2}\|\mathcal{G} - \mathcal{A}\|^2 + \gamma\|\mathcal{G}\|_{LTR}$ can be rewritten as:

$$
\begin{aligned}
&\frac{1}{2}\|\mathcal{G} - \mathcal{A}\|_F^2 + \gamma\|\mathcal{G}\|_{LTR} \\
&= \frac{1}{2n_3}\|\mathcal{G}_f - \mathcal{A}_f\|_F^2 + \frac{\gamma}{n_3}\sum_{k=1}^{n_3}\|\mathcal{G}_f^k\|_{LTR} \\
&= \frac{1}{n_3}\sum_{k=1}^{n_3}\left(\frac{1}{2}\|\mathcal{G}_f^k - \mathcal{A}_f^k\|_F^2 + \gamma\|\mathcal{G}_f^k\|_{LTR}\right).
\end{aligned}
\qquad (15)
$$

Thus, the original tensor optimization problem can be decoupled into $n_3$ independent matrix optimization problems:

$$\arg\min_{\mathcal{G}_f^k} \frac{1}{2}\|\mathcal{G}_f^k - \mathcal{A}_f^k\|_F^2 + \gamma\|\mathcal{G}_f^k\|_{LTR}. \qquad (16)$$

Here, $1 \leq k \leq n_3$, the SVD of $\mathcal{A}^k$ is $\mathcal{U}_f^k \mathcal{S}_f^k (\mathcal{V}_f^k)^H$. Since unitary transformations do not change the singular values, LTR functions that applied to singular values are evidently unitarily invariant functions. According to (Kang et al., 2015), the optimal solution of Eq. (16) is:

$$\mathcal{G}_f^{*k} = \mathcal{U}_f^k Prox_{f,\gamma}(\mathcal{S}_f^k)(\mathcal{V}_f^k)^H, \qquad (17)$$

where $Prox_{f,\gamma}(\mathcal{S}_f^k(i,i)) = \arg\min_{x\geq 0}\frac{1}{2}(x - \mathcal{S}_f^k(i,i))^2 + \gamma f(x)$ and $f(x) = 1 - \exp\left(-\frac{e^\delta x}{\delta}\right)$. $\qquad\square$

Given that Eq. (14) in **Theorem 3.1** is a combination of concave and convex functions, we can utilize DC Programming (Tao & An, 1997) to derive a closed-form solution:

$$\tau^{iter+1} = \left(\mathcal{S}_f^k(i,i) - \frac{\beta \cdot \partial f(\tau^{iter})}{\rho}\right)_+, \qquad (18)$$

---

**Algorithm 1** The Training Process of **TMvML**

**Input:**
  $\{\mathbf{X}^v\}_{v=1}^V$: Training examples;
  $\mathbf{Y}$: Label matrix;
  $\alpha$ and $\beta$: The trade-off parameters;
  $t$ : Number of iterations;
  $\varepsilon$ : Convergence threshold.
**Output:**
  $\hat{\mathbf{Y}}$: Predicted label matrix.
1: Initialize $\mathbf{W}^v$, $\mathbf{E}^v$, $\mathbf{A}^v$, $\mathbf{Z}^v$, $\mathbf{B}^v$, $\mathcal{G}$, $\mathcal{C}$, $\mu$ and $\rho$;
3: Compute the loss $\mathcal{L}_0$ by Eq. (4);
4: **for** $i = 1, 2, \ldots, t$ **do**
5:   **for** $v = 1, 2, \ldots, V$ **do**
6:     Optimize $\mathbf{Z}^v$ by Eq. (7) and Eq. (8);
7:     Optimize $\mathbf{E}^v$ by Eq. (10);
8:     Optimize $\mathbf{W}^v$ by Eq. (19);
9:     Optimize $\mathbf{A}^v$ by Eq. (20);
10:     Optimize $\mathbf{B}^v$ by Eq. (21);
11:   **end for**
12:   Optimize $\mathcal{G}$ by Eq. (18);
13:   Optimize $\mathcal{C}$, $\mu$ and $\rho$ by Eq. (21);
14:   Optimize $\mathcal{L}_i$ by Eq. (4);
15:   **if** $|\mathcal{L}_i - \mathcal{L}_{i-1}| \leq \varepsilon$ **break**;
16: **end for**

---

where $\tau = Prox_{f,\gamma}(\mathcal{S}_f^k(i,i))$, and $iter$ is the iterations.

**$\mathbf{W}^v$-subproblem:** With other variables fixed and taking the derivative of Eq. (5) with respect to $\mathbf{W}^v$ to zero, we have:

$$
\begin{aligned}
\mathbf{W}^v =& (2(\mathbf{Z}^v)^\top \mathbf{S}^\top \mathbf{S}\mathbf{Z}^v + \rho\mathbf{I})^{-1}(-\mathbf{C}^v \\
&+ 2(\mathbf{Z}^v)^\top \mathbf{S}^\top \mathbf{S}\mathbf{Y} + \rho\mathbf{G}^v).
\end{aligned}
\qquad (19)
$$

**$\mathbf{A}^v$-subproblem:** With other variables fixed, the subproblem for $\mathbf{A}^v$ is formulated as

$$\mathbf{A}^{v*} = \arg\min_{\mathbf{A}^v(\mathbf{A}^v)^\top = \mathbf{I}} \mathrm{Tr}((\mathbf{A}^v)^\top \mathbf{M}^v), \qquad (20)$$

where $\mathbf{M}^v = (\mathbf{Z}^v)^\top(\mu\mathbf{X}^v + \mathbf{B}^v - \mu\mathbf{E}^v)$. The optimal solution of $\mathbf{A}^v$ is $\mathbf{U}^v(\mathbf{V}^v)^\top$, where $\mathbf{U}^v$ and $\mathbf{V}^v$ are the left and right singular matrix of $\mathbf{M}^v$. Finally, the Lagrange multipliers and penalty parameters are updated as follows,

$$
\begin{cases}
\mathbf{B}^v = \mathbf{B}^v + \mu(\mathbf{X}^v - \mathbf{Z}^v\mathbf{A}^v - \mathbf{E}^v), \\
\mathcal{C} = \mathcal{C} + \rho(\mathcal{W} - \mathcal{G}), \\
\mu = \eta_\mu\mu, \ \rho = \eta_\rho\rho,
\end{cases}
\qquad (21)
$$

where $\eta_\mu, \eta_\rho > 1$ are used to accelerate convergence.

During the whole model training process, we first initialize the required variables, and then repeat the above steps until the algorithm converges or reaches the maximum iterations. Finally, we make prediction for unseen instances according to $\hat{\mathbf{Y}} = \frac{1}{V}\sum_{v=1}^V \mathbf{Z}_u^v\mathbf{W}^v$. Algorithm 1 summarizes the whole procedure of our proposed TMvML method.

### 3.4. Computation Complexity Analysis

The computation complexity of TMvML is mainly determined by the solution of its five subproblems in section 3.3, where the time complexity spent on $\{\mathbf{Z}^v, \mathbf{E}^v, \mathcal{G}, \mathbf{A}^v, \mathbf{W}^v\}$ are $\mathcal{O}(nk^2+nkd+nqk)$, $\mathcal{O}(nd)$, $\mathcal{O}(dqV\log(Vd)+qV^2d)$, $\mathcal{O}(nk^2+nkq)$, $\mathcal{O}(k^2(d)^2)$, respectively. $d$ is the maximum dimensionality of the multi-view data. In our experiments, since $n \gg q$, $n \gg k$, $n \gg d_{\max}$, the overall computation complexity of TMvML is $O(tnk^2)$, where $t$ is the number of iterations and usually no more than 50 on all datasets.

## 4. Experiments

### 4.1. Experimental Setting

To validate the effectiveness of TMvML, we conducted in-depth experimental analysis on six widely-used MVML datasets, including *Emotions*, *Yeast*, *Corel5k*, *Plant*, *Human* and *Espgame*, which can be downloaded from Mulan website: *http://mulan.sourceforge.net/datasets-mlc.html*. Table 1 summarizes the detailed characteristics of these datasets.

*Table 1.* The characteristics of our employed datasets: the number of samples ($n$), views ($v$), classes ($q$) and the maximum / minimum dimension of all views ($d_{\max}, d_{\min}$).

| Datasets | $n$ | $v$ | $q$ | $d_{\max}$ | $d_{\min}$ |
|---|---|---|---|---|---|
| Emotions | 593 | 2 | 6 | 64 | 8 |
| Plant | 978 | 2 | 12 | 400 | 40 |
| Yeast | 2417 | 2 | 14 | 79 | 24 |
| Human | 3106 | 2 | 14 | 400 | 40 |
| Corel5k | 4999 | 6 | 260 | 4096 | 100 |
| Espgame | 20770 | 6 | 268 | 4096 | 100 |

For comparative studies, we employ six state-of-the-art MVML methods, including LrMMC (Liu et al., 2015), SIMM (Wu et al., 2019), FIMAN (Wu et al., 2020), ICM2L (Tan et al., 2021), ML-BVAE (Fu et al., 2024) and IMvMLC (Wen et al., 2024), with parameters configured as recommended in their respective literature.

In addition, five widely-used evaluation metrics in multi-label learning are employed to measure the performance of algorithms, including *Average Precision* (AP), *Hamming Loss* (HL), *One Error* (OE), *Ranking Loss* (RL) and *Coverage* (COV), where the formal definition of the above metrics can be found in (Gibaja & Ventura, 2015). For each dataset, we randomly selected 70% data for training, 10% data for validation and 20% data for testing.

### 4.2. Experimental Results

To ensure reliable comparisons, we run each algorithm five times and record the average metric results and standard de-

viation in Table 2, with the best performances highlighted in bold. From the 180 comparisons (6 datasets × 6 comparing methods × 5 metrics), we can observe that:

- Among the six employed datasets, TMvML outperforms all comparing methods on *Emotions*, *Yeast*, *Plant* and *Human* datasets. And it is also superior to other methods on *Espgame* and *Corel5k* datasets in 86.67% and 96.67% cases, respectively.

- Among the five evaluation metrics, TMvML achieves the best performance on *Average Precision*, *One Error* and *Coverage* metrics in all cases. And on *Ranking Loss* and *Hamming Loss* metrics, it also superior to other methods in 94.44% and 91.67% cases.

- Compared with non-tensor methods (such as ICM2L), TMvML performs significant superiority on all employed datasets, which is attributed to its employed tensor architecture that can leverage the low-rank tensor property to capture higher-order correlations across views and labels.

- Compared with methods that employ independent components to mine view and label relationships, TMvML exhibits excellent performance on all datasets. We attribute such success to our designed tensor classifier, which can simultaneously excavate cross-view high-order correlations and characterize multi-label semantic relationships.

To comprehensively evaluate the superiority of TMvML, the *Friedman test* (Demšar, 2006) is utilized as the statistical test to determine whether multiple algorithms have the same performance. Table 3 shows the Friedman statistical $\tau_F$ value for each evaluation metric and the critical value. According to Table 3, all evaluation metrics reject the null hypothesis that "all algorithms perform equally" at a 0.05 significance level. Thus, we choose the post-hoc Bonferroni-Dunn test (Demšar, 2006) to further illustrate the differences among these methods. Figure 4 illustrates the CD diagrams for each evaluation metric, where the average rank of each algorithm is marked along the axis. According to Figure 4, it is clearly observed that our proposed TMvML consistently ranks 1st across all evaluation metrics.

## 5. Further Analysis

### 5.1. Ablation Study

To evaluate the contribution of our proposed Laplace Tensor Rank (LTR), we conducted additional ablation studies. Specifically, we compared our proposed TMvML with two variants where LTR in Eq. (4) was replaced by the state-of-the-art Enhanced Tensor Rank (ETR) (Ji & Feng,

*Table 2.* Experimental results (mean±std). ↑ represents the higher the better; ↓ represents the lower the better.

| | metrics | LrMMC | SIMM | FIMAN | ML-BVAE | IMvMLC | ICM2L | TMVML |
|---|---|---|---|---|---|---|---|---|
| **Emotions** | AP ↑ | 0.763±0.020 | 0.634±0.043 | 0.806±0.027 | 0.572±0.022 | 0.782±0.021 | 0.567±0.004 | **0.811±0.020** |
| | HL ↓ | 0.216±0.011 | 0.307±0.004 | 0.231±0.013 | 0.317±0.012 | 0.330±0.021 | 0.342±0.007 | **0.210±0.016** |
| | OE ↓ | 0.338±0.032 | 0.501±0.092 | 0.258±0.042 | 0.568±0.029 | 0.303±0.032 | 0.578±0.016 | **0.248±0.041** |
| | RL ↓ | 0.161±0.016 | 0.344±0.047 | 0.161±0.026 | 0.423±0.035 | 0.183±0.019 | 0.432±0.004 | **0.160±0.016** |
| | COV ↓ | 2.198±0.094 | 0.457±0.051 | 7.796±0.189 | 3.163±0.242 | 1.873±0.037 | 3.091±0.027 | **0.300±0.070** |
| | metrics | LrMMC | SIMM | FIMAN | ML-BVAE | IMvMLC | ICM2L | TMVML |
| **Yeast** | AP ↑ | 0.610±0.013 | 0.712±0.014 | 0.740±0.007 | 0.712±0.007 | 0.738±0.006 | 0.695±0.001 | **0.771±0.008** |
| | HL ↓ | 0.255±0.020 | 0.241±0.011 | 0.216±0.004 | 0.232±0.004 | 0.313±0.007 | 0.231±0.001 | **0.208±0.005** |
| | OE ↓ | 0.309±0.028 | 0.253±0.021 | 0.257±0.013 | 0.253±0.012 | 0.248±0.009 | 0.264±0.001 | **0.215±0.007** |
| | RL ↓ | 0.275±0.011 | 0.218±0.018 | 0.187±0.005 | 0.204±0.007 | 0.180±0.005 | 0.213±0.001 | **0.167±0.007** |
| | COV ↓ | 10.32±0.195 | 7.368±0.293 | 6.673±0.074 | 6.706±0.094 | 6.538±0.094 | 6.687±0.031 | **0.470±0.002** |
| | metrics | LrMMC | SIMM | FIMAN | ML-BVAE | IMvMLC | ICM2L | TMVML |
| **Corel5k** | AP ↑ | 0.215±0.010 | 0.292±0.004 | 0.430±0.007 | 0.286±0.000 | 0.333±0.008 | 0.210±0.000 | **0.440±0.008** |
| | HL ↓ | 0.013±0.000 | 0.013±0.018 | 0.018±0.000 | 0.013±0.000 | 0.158±0.008 | 0.020±0.000 | **0.013±0.000** |
| | OE ↓ | 0.776±0.015 | 0.614±0.012 | 0.489±0.017 | 0.626±0.008 | 0.613±0.019 | 0.797±0.000 | **0.476±0.013** |
| | RL ↓ | 0.173±0.004 | 0.160±0.005 | **0.085±0.000** | 0.188±0.008 | 0.114±0.003 | 0.174±0.001 | 0.108±0.003 |
| | COV ↓ | 96.72±1.300 | 95.99±3.146 | 53.94±0.790 | 108.7±4.792 | 70.80±2.256 | 98.383±0.283 | **0.266±0.006** |
| | metrics | LrMMC | SIMM | FIMAN | ML-BVAE | IMvMLC | ICM2L | TMVML |
| **Plant** | AP ↑ | 0.464±0.016 | 0.369±0.029 | 0.492±0.030 | 0.505±0.020 | 0.544±0.017 | 0.587±0.047 | **0.608±0.007** |
| | HL ↓ | 0.115±0.002 | 0.090±0.001 | 0.238±0.011 | 0.090±0.001 | 0.202±0.038 | 0.920±0.006 | **0.087±0.004** |
| | OE ↓ | 0.668±0.018 | 0.809±0.037 | 0.696±0.031 | 0.705±0.032 | 0.652±0.027 | 0.637±0.064 | **0.570±0.006** |
| | RL ↓ | 0.371±0.014 | 0.378±0.025 | 0.277±0.028 | 0.238±0.012 | 0.210±0.015 | 0.853±0.027 | **0.168±0.015** |
| | COV ↓ | 4.256±0.147 | 4.325±0.270 | 3.216±0.350 | 2.749±0.141 | 2.461±0.171 | 1.715±0.317 | **0.169±0.013** |
| | metrics | LrMMC | SIMM | FIMAN | ML-BVAE | IMvMLC | ICM2L | TMVML |
| **Espgame** | AP ↑ | 0.170±0.003 | 0.304±0.002 | 0.284±0.002 | 0.257±0.001 | 0.273±0.001 | 0.226±0.000 | **0.306±0.001** |
| | HL ↓ | 0.028±0.000 | 0.017±0.000 | 0.028±0.000 | **0.017±0.000** | 0.021±0.002 | 0.027±0.000 | 0.026±0.000 |
| | OE ↓ | 0.992±0.001 | 0.536±0.004 | 0.628±0.004 | 0.615±0.010 | 0.615±0.013 | 0.692±0.000 | **0.506±0.002** |
| | RL ↓ | 0.410±0.003 | 0.164±0.003 | 0.154±0.002 | 0.211±0.002 | **0.141±0.000** | 0.207±0.000 | 0.141±0.002 |
| | COV ↓ | 210.9±1.020 | 110.1±1.260 | 102.8±1.183 | 136.5 ±1.681 | 92.21±0.113 | 143.124±0.012 | **0.409±0.008** |
| | metrics | LrMMC | SIMM | FIMAN | ML-BVAE | IMvMLC | ICM2L | TMVML |
| **Human** | AP ↑ | 0.480±0.006 | 0.495±0.042 | 0.583±0.015 | 0.536±0.010 | 0.600±0.010 | 0.603±0.034 | **0.631±0.010** |
| | HL ↓ | 0.096±0.002 | 0.085±0.001 | 0.151±0.002 | 0.085±0.001 | 0.112±0.006 | 0.920±0.002 | **0.083±0.003** |
| | OE ↓ | 0.673±0.009 | 0.663±0.034 | 0.585±0.020 | 0.659±0.014 | 0.578±0.015 | 0.595±0.054 | **0.542±0.016** |
| | RL ↓ | 0.358±0.006 | 0.261±0.046 | 0.186±0.011 | 0.181±0.008 | 0.149±0.007 | 0.879±0.016 | **0.136±0.004** |
| | COV ↓ | 5.281±0.072 | 3.797±0.640 | 2.817±0.095 | 2.666±0.112 | 2.271±0.099 | 1.832±0.211 | **0.150±0.003** |

*Table 3.* Friedman statics $\tau_F$ of each evaluation metric.

| Evaluation Metric | $\tau_F$ | critical value ($\alpha$=0.05) |
|---|---|---|
| Average Precision | 8.696 | |
| Hamming Loss | 4.805 | 2.421 |
| One Error | 7.714 | |
| Ranking Loss | 13.129 | Methods: 7, Data sets: 6 |
| Coverage | 7.600 | |

2025), and the traditional Tensor Nuclear Norm (TNN) (Xie et al., 2018), denoted as TMvML-ETR and TMvML-TNN respectively. As shown in Figure 5, when varying the parameter $\delta$ across the search range of $\{10^{-4}, \cdots, 1\}$, our TMvML method consistently outperforms TMvML-ETR and TMvML-TNN in most cases and achieves the best classification performance in the optimal setting. This phenomenon is attributed to the distinct ways these methods treat singular values. While LTR and ETR apply varying penalties to individual singular values, TNN uniformly scales them all, thus overlooking differentiated information between large and small singular values in the tensor data. Notably, compared to ETR, LTR preserves larger singular values to 1, effectively characterizing the low-rank tensor structure. Additionally, since the parameter $\delta$ affects the different constraints in varying ways, TMvML-ETR may outperform TMvML in a few cases.

In addition, we further analyze the effect of the parameters $\delta$ on the classification results. As shown in Figure 5, we can observe that $\delta$ has a significant effect. The best classification

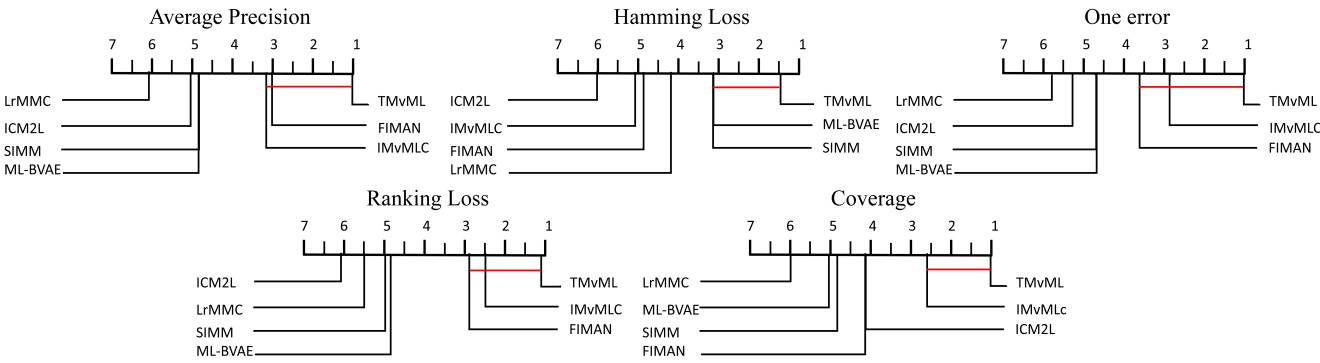

*Figure 4.* Experimental comparisons between our proposed TMvML method and all other comparing algorithms on five evaluation metrics with the Bonferroni-Dunn test (CD = 3.213 at 0.05 significance level).

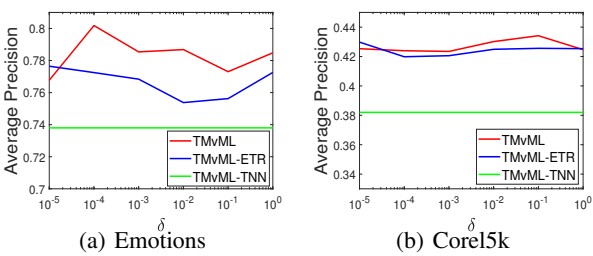

*Figure 5.* The Average Precision of TMvML and TMvML-ETR with varying $\delta$ on *Emotions* and *Corel5k* datasets.

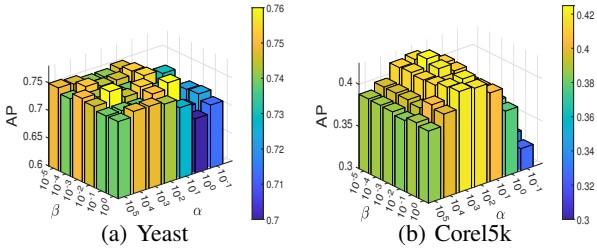

*Figure 6.* The parameter sensitivity analysis of TMvML under different $\alpha$ and $\beta$ configurations on *Yeast* and *Corel5k* datasets.

results for *Emotions* are obtained when $\delta = 10^{-4}$, while *Corel5k* peaks at $\delta = 10^{-1}$. The main reasons of such phenomenon lie in that, since $\delta$ determines the strength of the penalty for singular values, and the distribution of singular values varies from one dataset to another, each dataset needs an appropriate parameter to provide better discriminability power for the learned low-rank representation tensor.

### 5.2. Parameter Sensitivity

We study the sensitivity analysis of our proposed TMvML with respect to its two key hyperparameters $\alpha$ and $\beta$. We

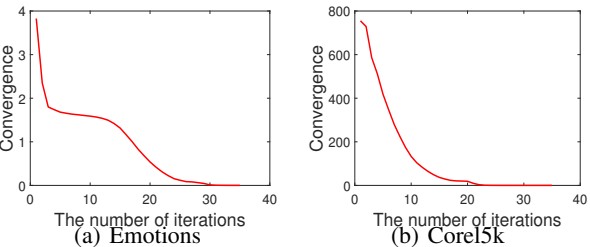

*Figure 7.* Convergence analysis of TMvML on *Emotions* and *Corel5k* datasets.

change the value of $\alpha$ within the set of $\{10^{-1}, 10^0, \dots, 10^5\}$ and $\beta$ within the set of $\{10^{-5}, 10^{-4}, \dots, 10^0\}$. According to Figure 6, the performance of TMvML is stable when hyperparameter $\alpha$ is around $10^2$ to $10^5$ and hyperparameter $\beta$ is around $10^{-5}$ to $10^0$. Such phenomenon illustrates that the performance of our proposed TMvML is stable across a broad range of parameters, which also empirically demonstrates its efficiency and robustness.

### 5.3. Convergence Analysis

We further analyze the convergence of the our proposed TMvML. Figure 7 shows the convergence curves of TMvML method on the *Emotions* and *Corel5k* datasets. According to Figure 7, the value of the objective function drops sharply at the beginning of the iterations and gradually stabilizes as the number of iterations increases. Similar convergence results are also observed in other datasets, which empirically verifies the convergence of TMvML.

## 6. Conclusion

In this paper, we proposed a Tensorized Multi-View Multi-Label Classification method via Laplace Tensor Rank

(TMvML), which is the first attempt to utilize tensorized low-rank MVML classifier to achieve the high-order feature correlations extraction and multi-label semantic correlations characterization simultaneously. To better characterize the tensor low-rank structure, we designed a new Laplace Tensor Rank (LTR), which serves as a tighter surrogate of tensor rank to effectively capturing high-order fiber correlations. Extensive results demonstrate that our proposed TMvML is significantly superior to other state-of-the-art methods.

## Impact Statement

This paper presents work whose goal is to advance the field of Machine Learning. There are many potential societal consequences of our work, none which we feel must be specifically highlighted here.

## Acknowledgments

This work was supported by the National Natural Science Foundation of China (No. 62306020), the Young Elite Scientist Sponsorship Program by BAST (No. BYESS2024199), Beijing Natural Science Foundation (No. L244009), the National College Students' Innovation and Entrepreneurship Training Program of BJUT (No. GJDC2025-01-32), the Major Program of the National Social Science Foundation of China (No. 22&ZD147), and the National Key Research and Development Program of China (No. 2023YFB3107100).

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

## A. Proofs of Theorem 3.2

**Theorem A.1.** *Let $\{\mathcal{P}_k = (\mathbf{Z}_k^v, \mathbf{E}_k^v, \mathbf{A}_k^v, \mathbf{W}_k^v, \mathbf{B}_k^v, \mathcal{C}_k, \mathcal{G}_k)\}_{k=0}^{\infty}$ be the sequence generated by Algorithm 1, then the sequence $\{P_k\}$ meets the following two principles:*

*1). $\{P_k\}$ is bounded;*

*2). Any accumulation point of $\{P_k\}$ is a KKT point of Algorithm 1.*

To prove Theorem 3.2, we first introduce two important lemmas.

**Lemma A.2.** *(Lin et al., 2010) Let $\mathcal{H}$ be a real Hilbert space endowed with an inner product $\langle \cdot, \cdot \rangle$, a norm $\| \cdot \|$ with the dual norm $\| \cdot \|^{dual}$, and $y \in \partial \|x\|$, where $\partial f(x)$ is the subgradient of $f(\cdot)$. Then we have $\|y\|^{dual} = 1$ if $x \neq 0$, and $\|y\|^{dual} \leq 1$ if $x = 0$.*

**Lemma A.3.** *(Lewis & Sendov, 2005) Suppose that $F : \mathbb{R}^{m \times n} \to \mathbb{R}$ is defined as $F(\mathbf{X}) = f \circ \sigma(\mathbf{X}) = f(\sigma_1(\mathbf{X}), \ldots, \sigma_r(\mathbf{X}))$, where $\mathbf{X} = \mathbf{U}Diag(\sigma(\mathbf{X}))\mathbf{V}^T$ is the SVD of matrix $\mathbf{X} \in \mathbb{R}^{m \times n}$, $r = \min(m, n)$, and $f : \mathbb{R}^r \to \mathbb{R}$ is differentiable and absolutely symmetric at $\sigma(\mathbf{X})$. Then, the subdifferential of $F(\mathbf{X})$ at $\mathbf{X}$ is*

$$\frac{\partial F(\mathbf{X})}{\partial \mathbf{X}} = \mathbf{U}Diag(\partial f(\sigma(\mathbf{X})))\mathbf{V}^T,$$

*where*

$$\partial f(\sigma(\mathbf{X})) = (\frac{\partial f(\sigma_1(x))}{\partial \mathbf{X}}, \ldots, \frac{\partial f(\sigma_r(x))}{\partial \mathbf{X}}).$$

*Proof.* Proof of the first part: On the $k + 1$ iteration, from the updating rule of $\mathbf{E}_{k+1}$, the first-order optimal condition should be satisfied.

$$
\begin{aligned}
0 &= \alpha\partial \left\| \mathbf{E}_{k+1}^v \right\|_{2,1} \\
&\quad + \mu_k(\mathbf{E}_{k+1}^v - (\mathbf{X}^v - \mathbf{Z_{k+1}}^v \mathbf{A}^v + \mathbf{B}_k^v/\mu_k)) \\
&= \alpha\partial \left\| \mathbf{E}_{k+1}^v \right\|_{2,1} - \mathbf{B}_{k+1}^v,
\end{aligned}
\tag{22}
$$

Thus, we have

$$\frac{1}{\alpha}[\mathbf{B}_{k+1}^v]_{i,j} = \partial \left\| \left[\mathbf{E}_{k+1}^v\right]_{:,j} \right\|_2,$$

where $[\mathbf{B}_{k+1}^v]_{i,j}$ and $[\mathbf{E}_{k+1}^v]_{i,j}$ are the $j$-th columns of $\mathbf{B}_{k+1}^v$ and $\mathbf{E}_{k+1}^v$. And the $\ell_2$ norm is self-dual, so based on the Lemma A.2, we have $\left\| \frac{1}{\alpha}[\mathbf{B}_{k+1}^v]_{:,j} \right\|_2 \geq 1$. So the sequence $\{\mathbf{B}_{k+1}^v\}$ is bounded.

Then, we prove the sequence $\{\mathcal{C}_{k+1}\}$ is bounded. According to the updating rule of $\mathcal{G}$, the first-order optimality condition holds

$$\partial \left\| \mathcal{G}_{k+1} \right\|_{\text{LTR}} = \mathcal{C}_{t+1}.
\tag{23}$$

Let $\mathcal{U} * \mathcal{S} * \mathcal{V}^T$ be the t-SVD of tensor $\mathcal{G}$. According to the Lemma A.3 and definition of LTR, we have:

$$\left\| \partial \left\| \mathcal{G}_{k+1} \right\|_{\mathrm{LTR}} \right\|_F^2$$

$$= \left\| \frac{1}{n_3} \mathcal{U} * ifft(\partial(\mathcal{S}_f), [], 3) * \mathcal{V}^T \right\|_F^2$$

$$= \frac{1}{n_3^3} \left\| \partial f(\mathcal{S}_f) \right\|_F^2 \tag{24}$$

$$\leq \frac{1}{n_3^3} \sum_{i=1}^{n_3} \sum_{j=1}^{min(n_1,n_2)} \left[ \partial f(\mathcal{S}_f^v(j,j)) \right]^2$$

$$\leq \frac{e^{2\delta} min(n_1, n_2)}{\delta^2 n_3^2}$$

where the second inequality is by the fact $\partial f(x) \leq \frac{e^\delta}{\delta}$, and $f_{\mathrm{LTR}}(x) = 1 - \exp\left( -\frac{e^\delta x}{\delta} \right)$ is our rank approximation function. So $\partial \left\| \mathcal{G}_{k+1} \right\|_{\mathrm{LTR}}$ is bounded, meanwhile the sequence $\{ \mathcal{C}_{k+1} \}$ is also bounded.

Moreover, from the iterative method in the algorithm of solving TMvML, we can deduce

$$\mathcal{L}_{\mu_k,\rho_k}(\mathbf{Z}_{k+1}^v, \mathbf{W}_{k+1}^v, \mathbf{E}_{k+1}^v, \mathbf{A}_{k+1}^v, \mathcal{G}_{k+1}, \mathbf{B}_k^v, \mathcal{C}_k)$$

$$\leq \mathcal{L}_{\mu_k,\rho_k}(\mathbf{Z}_k^v, \mathbf{W}_k^v, \mathbf{E}_k^v, \mathbf{A}_k^v, \mathcal{G}_k, \mathbf{B}_k^v, \mathcal{C}_k)$$

$$= \mathcal{L}_{\mu_{k-1},\rho_{k-1}}(\mathbf{Z}_k^v, \mathbf{W}_k^v, \mathbf{E}_k^v, \mathbf{A}_k^v, \mathcal{G}_k, \mathbf{B}_{k-1}^v, \mathcal{C}_{k-1})$$

$$+ \frac{\rho_k + \rho_{k-1}}{2\rho_{k-1}^2} \left\| \mathcal{C}_k - \mathcal{C}_{k-1} \right\|_F^2 \tag{25}$$

$$+ \frac{\mu_k + \mu_{k-1}}{2\mu_{k-1}^2} \sum_{v=1}^V \left\| \mathbf{B}_k^v - \mathbf{B}_{k-1}^v \right\|_F^2,$$

Thus, summing two sides of Eq.(25) form $k = 1$ to $n$,

$$\mathcal{L}_{\mu_k,\rho_k}(\mathbf{Z}_{k+1}^v, \mathbf{W}_{k+1}^v, \mathbf{E}_{k+1}^v, \mathbf{A}_{k+1}^v, \mathcal{G}_{k+1}, \mathbf{B}_k^v, \mathcal{C}_k)$$

$$\leq \mathcal{L}_{\mu_0,\rho_0}(\mathbf{Z}_1^v, \mathbf{W}_1^v, \mathbf{E}_1^v, \mathbf{A}_1^v, \mathcal{G}_1, \mathbf{B}_0^v, \mathcal{C}_0)$$

$$+ \sum_{k=1}^n \frac{\rho_k + \rho_{k-1}}{2\rho_{k-1}^2} \left\| \mathcal{C}_k - \mathcal{C}_{k-1} \right\|_F^2 \tag{26}$$

$$+ \sum_{k=1}^n \left( \frac{\mu_k + \mu_{k-1}}{2\mu_{k-1}^2} \sum_{v=1}^V \left\| \mathbf{B}_k^v - \mathbf{B}_{k-1}^v \right\|_F^2 \right)$$

Observe that

$$\sum_{k=1}^n \frac{\mu_k + \mu_{k+1}}{2\mu_{k-1}^2} < \infty, \quad \sum_{k=1}^n \frac{\rho_k + \rho_{k+1}}{2\rho_{k-1}^2} < \infty \tag{27}$$

Note that $\mathcal{L}_{\mu_0,\rho_0}(\mathbf{Z}_1^v, \mathbf{W}_1^v, \mathbf{E}_1^v, \mathbf{A}_1^v, \mathcal{G}_1, \mathbf{B}_0^v, \mathcal{C}_0)$ is finite, and sequence $\{\mathbf{B}_k^v\}$, $\{\mathcal{C}_k\}$, $\sum_{k=1}^n \frac{\mu_k + \mu_{k+1}}{2\mu_{k-1}^2}$ and $\sum_{k=1}^n \frac{\rho_k + \rho_{k+1}}{2\rho_{k-1}^2}$ are all bounded. So $\mathcal{L}_{\mu_k,\rho_k}(\mathbf{Z}_{k+1}^v, \mathbf{W}_{k+1}^v, \mathbf{E}_{k+1}^v, \mathbf{A}_{k+1}^v, \mathcal{G}_{k+1}, \mathbf{B}_k^v, \mathcal{C}_k)$ is bounded.

Notice

$$\mathcal{L}_{\mu_k, \rho_k}(\mathbf{Z}_{k+1}^v, \mathbf{W}_{k+1}^v, \mathbf{E}_{k+1}^v, \mathbf{A}_{k+1}^v, \mathcal{G}_{k+1}, \mathbf{B}_k^v, \mathcal{C}_k)$$

$$= \alpha \left\| \mathbf{E}_{k+1} \right\|_{2,1} + \beta \|\mathcal{G}_{k+1}\|_{\text{LTR}} + \sum_{v=1}^{V} \left\| \mathbf{S} \left( \mathbf{Y} - \mathbf{Z}_{k+1}^v \mathbf{W}_{k+1}^v \right) \right\|_{\text{F}}^2$$

$$+ \sum_{v=1}^{V} \left( (\mathbf{B}_k^v, \mathbf{X}^v - \mathbf{Z}_{k+1}^v \mathbf{A}_{k+1}^v - \mathbf{E}_{k+1}^v) + \frac{\mu_k}{2} \left\| \mathbf{X}^v - \mathbf{Z}_{k+1}^v \mathbf{A}_{k+1}^v - \mathbf{E}_{k+1}^v \right\|_F^2 \right)$$

$$+ \langle \mathcal{C}_k, \mathcal{W}_{k+1} - \mathcal{G}_{k+1} \rangle + \frac{\rho_k}{2} \left\| \mathcal{W}_{k+1} - \mathcal{G}_{k+1} \right\|_F^2, \tag{28}$$

and each term of Eq.(28) is nonnegative, due to the boundedness of $\mathcal{L}_{\mu_k, \rho_k}(\mathbf{Z}_{k+1}^v, \mathbf{W}_{k+1}^v, \mathbf{E}_{k+1}^v, \mathbf{A}_{k+1}^v, \mathcal{G}_{k+1}, \mathbf{B}_k^v, \mathcal{C}_k)$, we can deduce each term of Eq.(28) is bounded. So the boundedness of $\|\mathcal{G}_{k+1}\|_{\text{LTR}}$ implies that all singular values of $\mathcal{G}_{k+1}$ are bounded. Furthermore, based on the following equation

$$\|\mathcal{G}_{k+1}\|_F^2 = \frac{1}{n_3} \|\mathcal{G}_{f,k+1}\|_F^2 = \frac{1}{n_3} \sum_{i=1}^{n_3} \sum_{j=1}^{min(n_1,n_2)} \left( \mathcal{S}_f^i(j,j) \right)^2, \tag{29}$$

we can derive the sequence $\{\mathcal{G}_{k+1}\}$ is bounded, then, it is easy to prove the boundedness of $\{\mathbf{Z}_{k+1}\}$ and $\{\mathbf{A}_{k+1}\}$.

Therefore, we can conclude that the sequence $(\mathbf{Z}_k^v, \mathbf{E}_k^v, \mathbf{A}_k^v, \mathbf{W}_k^v, \mathbf{B}_k^v, \mathcal{C}_k, \mathcal{G}_k)_{k=0}^{\infty}$ generated by the Algorithm 1.

**2).** Proof of $2nd$ part: According to Weierstrass-Bolzano theorem (Bartle & Sherbert, 2000), there is at least one accumulation point of the sequence $\{\mathcal{P}_k\}_{k=1}^{\infty}$, we denote one of the points as $\mathcal{P}_*$. Then we have

$$\lim_{k \to \infty} (\mathbf{Z}_k^v, \mathbf{E}_k^v, \mathbf{A}_k^v, \mathbf{W}_k^v, \mathbf{B}_k^v, \mathcal{C}_k, \mathcal{G}_k) = (\mathbf{Z}_*^v, \mathbf{E}_*^v, \mathbf{A}_*^v, \mathbf{W}_*^v, \mathbf{B}_*^v, \mathcal{C}_*, \mathcal{G}_*).$$

Form the updating rule of $\mathcal{C}$ and $\mathbf{B}^v$, we have the following equations:

$$\mathbf{X}^v - \mathbf{Z}_{k+1}^v \mathbf{A}_{K+1}^v - \mathbf{E}_{k+1}^v = (\mathbf{B}_{k+1}^v - \mathbf{B}_k^v)/\mu_t,$$

$$\mathcal{W}_{k+1} - \mathcal{G}_{k+1} = (\mathcal{C}_{k+1} - \mathcal{C}_k)/\rho_t.$$

According the boundedness of sequences $\{\mathcal{C}_k\}$ and $\{\mathbf{B}_k^v\}$, and the fact $\lim_{k \to \infty} \mu_k = \infty$, we have:

$$\lim_{k \to \infty} \mathbf{X}^v - \mathbf{Z}_{k+1}^v \mathbf{A}_{K+1}^v - \mathbf{E}_{k+1}^v = \lim_{k \to \infty} (\mathbf{B}_{k+1}^v - \mathbf{B}_k^v)/\mu_t = 0,$$

$$\lim_{k \to \infty} \mathcal{W}_{k+1} - \mathcal{G}_{k+1} = \lim_{k \to \infty} (\mathcal{C}_{k+1} - \mathcal{C}_k)/\rho_t = 0,$$

then, we can obtain

$$\mathbf{X}^v - \mathbf{Z}_*^v \mathbf{A}_*^v - \mathbf{E}_*^v = 0, \quad \mathcal{W}_* - \mathcal{G}_* = 0.$$

Furthermore, due to the first-order optimality conditions of $\mathbf{E}_{k+1}^v$ and $\mathcal{G}_{k+1}$, we can deduce:

$$0 = \alpha \partial \left\| \mathbf{E}_{k+1}^v \right\|_{2,1} - \mathbf{B}_{k+1}^v \Rightarrow \mathbf{B}_*^v = \alpha \partial \left\| \mathbf{E}_*^v \right\|_{2,1}$$

$$0 = \beta \partial \left\| \mathcal{G}_{k+1} \right\|_{\text{LTR}} - \mathcal{C}_{k+1} \Rightarrow \mathcal{C}_* = \beta \partial \left\| \mathcal{G}_* \right\|_{\text{LTR}}$$

Thus, the accumulation point $\mathcal{P}_*$ of sequence $\{\mathcal{P}_k\}_{k=1}^{\infty}$ generated by the algorithm of solving TMvML satisfied the KKT condition. $\square$

# B. More Experimental Results

This section presents the experimental results for all six challenging datasets, including the ablation study of tensor rotation and Laplace Tensor Rank (LTR).

**Ablation Study of LTR**: Table (4) reports the results of ablation experiments of LTR conducted across six datasets. Specifically, table (4) compares TMvML with its variant TMvML-TNN, where the LTR was replaced by Tensor Nuclear Norm (TNN) (Xie et al., 2018) to capture the low-rank tensor structure. Experimental results demonstrate that TMvML consistently outperforms TMvML-TNN across all datasets. This compelling evidence proves that our proposed LTR is more effective than traditional TNN in modeling the complex high-order correlations in multi-view multi-label learning tasks.

*Table 4.* The Performance of TMvML and TMvML-TNN

|           |     | Emotions | Yeast | Corel5k | Plant | Espgame | Human |
|-----------|-----|----------|-------|---------|-------|---------|-------|
| TMvML     | AP  | **0.811** | **0.771** | **0.440** | **0.608** | **0.306** | **0.631** |
| TMvML-TNN | AP  | 0.738 | 0.747 | 0.382 | 0.601 | 0.270 | 0.621 |
| TMvML     | Cov | **0.300** | **0.460** | **0.266** | **0.169** | **0.409** | **0.150** |
| TMvML-TNN | Cov | 0.344 | 0.467 | 0.279 | 0.171 | 0.452 | 0.162 |

**Ablation Study of Tensor Rotation**: Table (5) reports the results of ablation experiments of tensor rotation conducted across six datasets. Specifically, Table (5) compared TMvML with a variant that removes the rotation operation (denoted as TMvML-NoRot). The results show that TMvML consistently outperforms TMvML-NoRot across all datasets. This significant performance gap highlights the critical role of rotation in extraction of both cross-view consistent correlations and multi-label semantic relationships simultaneously.

*Table 5.* The Performance of TMvML and TMvML-NoRot

|             |     | Emotions | Yeast | Corel5k | Plant | Espgame | Human |
|-------------|-----|----------|-------|---------|-------|---------|-------|
| TMvML       | AP  | **0.811** | **0.771** | **0.440** | **0.608** | **0.306** | **0.631** |
| TMvML-NoRot | AP  | 0.628 | 0.733 | 0.231 | 0.511 | 0.191 | 0.532 |
| TMvML       | Cov | **0.300** | **0.470** | **0.266** | **0.169** | **0.409** | **0.150** |
| TMvML-NoRot | Cov | 0.473 | 0.485 | 0.350 | 0.210 | 0.498 | 0.185 |

