# OpenReview forum: "Tensorized Multi-View Multi-Label Classification via Laplace Tensor Rank"
_ICML.cc/2025/Conference — ICML 2025 poster_

### Official Review · Reviewer_DSYf · 2025-02-22

**Overall Recommendation:** 5

**Summary:**

In this paper, the authors propose a novel approach that introduces a low-rank tensor classifier combined with the innovative Laplace Tensor Rank (LTR), which jointly captures high-order feature correlations and label dependencies. Extensive experiments across six benchmark datasets demonstrate TMvML’s superior performance.

**Claims And Evidence:**

Yes, the central claims are supported by evidence. Extensive experiments across six benchmark datasets demonstrate TMvML’s superior performance. The superiority of LTR over existing tensor rank approximations is validated through ablation studies.

**Essential References Not Discussed:**

All relevant works critical to understanding the main contribution of the method are cited in the paper.

**Experimental Designs Or Analyses:**

Yes, the experimental designs is reasonable. The use of six widely adopted MVML datasets and five standard metrics ensures a comprehensive and fair comparison. The inclusion of statistical tests further strengthens the reliability of the results.

**Methods And Evaluation Criteria:**

Yes, the methodological design is reasonable. The rotation operation on the tensor classifier is a clever design choice, enabling the exploration of interactions between different views and labels through frontal slice comparisons.

**Other Comments Or Suggestions:**

See the above weaknesses.

**Other Strengths And Weaknesses:**

Strengths:

(1) This work innovatively leverages a concise low-rank MVML tensor classifier to excavate cross-view feature correlations and characterize multi-label semantic relationships simultaneously. The whole paper is well organized and easy to understand.

(2) This paper designs a new Laplace Tensor Rank, which preserves larger singular values and discards smaller ones to obtain an accurate low-rank tensor representation. Such new component is welcome to multi-view community.

Weaknesses：

(1) The Figure 2 is ambiguous. Unclear labeling of vertical and horizontal coordinates, for example, why the true rank is 3 and the singular value gets progressively larger from 0 to 9.

(2) Some expressions are not accurate enough. For example, The matrix S is designed to ensure the multi-view representation is predictive corresponding to the known labels, and it is not clear.

**Questions For Authors:**

See the above weaknesses.

**Relation To Broader Scientific Literature:**

TMvML builds on prior work in tensor-based method and multi-label classification. The paper extends the principles of low-rank representation to the MVML setting, leveraging a novel tensorized classifier to capture high-order correlations across views and labels.

**Theoretical Claims:**

Yes, the proofs for Theorem 3.1 is correct.

---

> ### Author Rebuttal · Authors · 2025-03-29
>
> Thank you for the feedback on our paper. We appreciate the time and effort you have put into reviewing our work. In this rebuttal, we respond to the concerns raised in the reviews.
>
> **W1:** In this figure, we tested the ability of multiple low-rank tensor norms (including TNN, ETR, LTSpN and LTR) to approximate the true rank of three-dimensional tensors. Specifically, we constructed **a series of three-dimensional tensors with a fixed true rank of 3, while varying singular values that progressively increase from 0 to 9.** The horizontal axis represents the singular values, while the vertical axis represents the approximation value of the rank function. Such setup allows us to evaluate how well each method approximates the true rank under different singular value distributions.
>
> We will revise the figure to include a detailed caption explaining the experimental setup. We hope these changes address the reviewer’s concerns.
>
> **W2:** To clarify, our learning process follows a **transductive learning paradigm**, where the model leverages both labeled and unlabeled data during training but only uses the labels from the training set for supervision. The matrix $\bf S$ is a filtering matrix designed to ensure that the optimization process only utilizes the label information from the training set, while excluding any label information from the test set. Specifically, $\bf S$ is defined as a diagonal matrix:
>
> $\mathbf{S}_{ii}=\begin{cases}1&\text{if the }i\text{-th sample belongs to the training set,}\\\\0&\text{otherwise (samples belongs to the test set).}\end{cases}$

---

### Official Review · Reviewer_3DbN · 2025-03-02

**Overall Recommendation:** 3

**Summary:**

This paper proposes a method named TMvML for multi-view multi-label learning (MVML). The approach includes a low-rank tensor classifier to capture both consistent correlations across views and modeling complex multi-label relationships. Additionally, a new Laplace Tensor Rank (LTR) is introduced to capture higher-order correlations within the tensor space. This approach leads to significant improvements in MVML, as demonstrated by extensive experiments.

**Claims And Evidence:**

The paper's main claims are supported by convincing evidence. Extensive experiments on six datasets demonstrate TMvML’s superiority over state-of-the-art methods across multiple metrics.

**Essential References Not Discussed:**

There are no related works that are not currently discussed in the paper.

**Experimental Designs Or Analyses:**

I checked the validity of the experimental designs and analyses. Extensive experiments are conducted on six widely-used MVML benchmark datasets, with results averaged over multiple runs to ensure statistical reliability. The issues are listed behind in the Weaknesses.

**Methods And Evaluation Criteria:**

The proposed methods make sense for the problem. TMvML innovatively leverages tensor classifier to encode high-order correlations across both multi-view and multi-label, while the Laplace Tensor Rank (LTR) constraint effectively balances the preservation of critical semantic relationships and the suppression of noise. The experimental evaluation utilizes widely recognized MVML benchmark datasets and state-of-the-art baseline methods.

**Other Comments Or Suggestions:**

I would like to learn about the authors' response to the weaknesses listed above, which may give me a clearer perspective on the paper's contribution.

**Other Strengths And Weaknesses:**

The paper proposes a Tensorized Multi-View MultiLabel Classification method (TMvML), which is the first attempt to utilize tensorized low-rank MVML classifier to achieve the high-order feature correlations extraction and multi-label semantic correlations characterization simultaneously. Meanwhile, the paper designs a new Laplace Tensor Rank (LTR), which serves as a tighter surrogate of tensor rank to effectively capturing high-order fiber correlations.

There are also some weaknesses：

1. In tensor-based methods, the Tensor Nuclear Norm (TNN) is commonly used to capture the low-rank structure of tensors [1,2]. The proposed Laplace Tensor Rank (LTR) should be compared with TNN in the experiments to better highlight its effectiveness and potential advantages.

2. The proposed tensor classifier construction involves merging view-specific mapping matrices and rotating the resulting tensor to align label-view interactions. While this rotating design is theoretically motivated, ablation studies would strengthen the claim that rotation is essential for capturing label consistency and view correlations. Such experiments would provide concrete evidence of the rotation operation’s contribution to the method’s overall performance.

3. In Fig 6, the font of the coordinate axis should be enlarged.

[1] Zhao S, Wen J, Fei L, et al. Tensorized incomplete multi-view clustering with intrinsic graph completion[C] // Proceedings of the AAAI Conference on Artificial Intelligence. 2023, 37(9): 11327-11335.

[2] Zhang C, Li H, Lv W, et al. Enhanced tensor low-rank and sparse representation recovery for incomplete multi-view clustering[C] // Proceedings of the AAAI conference on artificial intelligence. 2023, 37(9): 11174-11182.

**Questions For Authors:**

I would like to learn about the authors' response to the weaknesses listed above, which may give me a clearer perspective on the paper's contribution.

**Relation To Broader Scientific Literature:**

The method TMvML is the first attempt to utilize tensorized low-rank MVML classifier in MVML problem.

**Theoretical Claims:**

I checked the correctness of the proofs for theoretical claims, including the theorems and proofs related to the effectiveness of LTR and closed-form solution in optimization.

---

> ### Author Rebuttal · Authors · 2025-03-29
>
> Thank you for the feedback on our paper. We appreciate the time and effort you have put into reviewing our work. In this rebuttal, we respond to the concerns raised in the reviews.
>
> **W1**: We agree that comparing the proposed Laplace Tensor Rank (LTR) with the widely used Tensor Nuclear Norm (TNN) is essential to highlight the advantages of our method. In fact, we have already performed a comparison of the LTR function with the TNN function and the results are shown in Fig. 3. According to the figure, LTR provides a tighter approximation to the true rank function compared to TNN, especially for larger singular values. Theoretically, LTR’s nonconvex formulation more aggressively suppresses small (noise-corrupted) singular values while preserving larger (signal-carrying) ones, leading to a more accurate low-rank representation. This property is not fully captured by TNN, which tends to over-penalize larger singular values due to its convex nature.
>
> However, we fully agree with the reviewer that comparing LTR with TNN in the experiments would better highlight its effectiveness. Thus, we compared TMvML with its variant TMvML-TNN, where the LTR was replaced by TNN to capture the low-rank tensor structure, and we report the results for the numerical experiments:
>
> |  |    | Emotions | Yeast | Corel5k | Plant| Espgame |Human|
> |-|-|-|-|-|-|-|-|
> |TMvML|AP| **0.811±0.020** | **0.771±0.008** | **0.440±0.008** | **0.608±0.007** | **0.306±0.001** | **0.631±0.010**|
> |TMvML-TNN|AP|0.738±0.014|0.747±0.014|0.382±0.015|0.601±0.017|0.270±0.020|0.621±0.007|
> |TMvML|Cov| **0.300±0.070** | **0.460±0.002** | **0.266±0.006** | **0.169±0.013** | **0.409±0.008** | **0.150±0.003** |
> |TMvML-TNN|Cov|0.344±0.032|0.467±0.009|0.279±0.003|0.171±0.012|0.452±0.009|0.162±0.005|
>
> Experimental results demonstrate that TMvML consistently outperforms TMvML-TNN across all datasets. This compelling evidence proves that our proposed LTR is more effective than traditional TNN in modeling the complex high-order correlations in multi-view multi-label learning tasks, particularly in preserving discriminative singular values while suppressing noise-corrupted ones.
>
>
>
>
> **W2**: We agree that ablation study to validate the rotation operation in the tensor classifier construction would provide concrete evidence of the rotation’s contribution to capturing label consistency and view correlations. We compared TMvML with a variant that removes the rotation operation (denoted as TMvML-NoRot), and the results are summarized below:
>
> |  |    | Emotions | Yeast | Corel5k | Plant| Espgame |Human|
> |-|-|-|-|-|-|-|-|
> |TMvML|AP| **0.811±0.020** | **0.771±0.008** | **0.440±0.008** | **0.608±0.007** | **0.306±0.001** | **0.631±0.010** |
> |TMvML-NoRot|AP|0.628±0.021|0.733±0.021|0.231±0.002|0.511±0.015|0.191±0.001|0.532±0.005|
> |TMvML |Cov| **0.300±0.070** | **0.470±0.002** | **0.266±0.006** | **0.169±0.013** | **0.409±0.008** | **0.150±0.003** |
> |TMvML-NoRot|Cov|0.473±0.005|0.485±0.004|0.350±0.002|0.210±0.007|0.498±0.000 |0.185±0.002|
>
> The results show that TMvML consistently outperforms TMvML-NoRot across all datasets. This significant performance gap highlights the critical role of rotation in extraction of both cross-view consistent correlations and multi-label semantic relationships simultaneously.
>
> **W3**: Thank you for pointing this out. We will enlarge the font of the coordinate axes to improve readability and ensure better clarity in the visualization.

---

### Official Review · Reviewer_d2CW · 2025-03-09

**Overall Recommendation:** 4

**Summary:**

This paper presented a method for Multi-View Multi-Label Learning (TMvML) which utilizes tensorized MVML classifier to achieve the high-order feature correlations extraction and multi-label semantic relationships characterization simultaneously. Moreover, a new Laplace Tensor Rank is designed to characterize a better low-rank tensor structure. Experiments show some good results.


## update after rebuttal
Thank you for your response. The new explanations and experimental results have strengthened the evaluation. After reading the authors' response, I would like to raise my rating to "Accept".

**Claims And Evidence:**

The claims are supported by evidence. TMvML’s superiority is evident in its consistent outperformance of existing methods.

**Essential References Not Discussed:**

All related works are cited or discussed in the paper.

**Experimental Designs Or Analyses:**

The experimental designs and analyses are reasonable, with thorough benchmark datasets of varying scales and complexities. The experiment also performs relevant ablation experiments, convergence analysis and hyperparametric analysis.

**Methods And Evaluation Criteria:**

The method design makes sense. Motivated by the fact that tensor can characterize the low-rank structure of multi-dimensional, tensorized MVML classifier can deal with MVML problem.

**Other Comments Or Suggestions:**

- Recent tensor-based MVML methods should be compared, which can judge whether TMvML’s gains stem from tensorization itself. A direct comparison with recent tensor-based MVML methods would provide clearer insights into the specific contributions of tensorization and help validate the effectiveness of the proposed framework.

- The modified Laplace function used in LTR introduces an additional exponential term $e^{\delta}$ compared with the original Laplace function. The authors need to elaborate on the specific advantages of this modification.

**Other Strengths And Weaknesses:**

**Strengths:**

- The organization of this article is reasonable and well-written.

- The proposed LTR offers a non-convex surrogate for tensor rank.

- Extensive experiments demonstrate TMvML’s superior performance.

**Weaknesses:**

- Recent tensor-based MVML methods should be compared, which can judge whether TMvML’s gains stem from tensorization itself. A direct comparison with recent tensor-based MVML methods would provide clearer insights into the specific contributions of tensorization and help validate the effectiveness of the proposed framework.

- The modified Laplace function used in LTR introduces an additional exponential term $e^{\delta}$ compared with the original Laplace function. The authors need to elaborate on the specific advantages of this modification.

- The paper lacks a dedicated convergence analysis with formal theoretical proofs. Especially, are there formal proofs or conditions ensuring convergence to a stationary point? Addressing these points would enhance the theoretical rigor of the paper.

**Questions For Authors:**

Are there formal proofs or conditions that ensure the convergence to a stationary point?

**Relation To Broader Scientific Literature:**

This approach aligns with recent efforts to enhance tensor rank approximations, but goes further by integrating multi-view and multi-label learning into a unified framework.

**Theoretical Claims:**

The proofs for theoretical claims in optimization are correct.

---

> ### Author Rebuttal · Authors · 2025-03-29
>
> Thank you for the feedback on our paper. We appreciate the time and effort you have put into reviewing our work. In this rebuttal, we respond to the concerns raised in the reviews.
>
> **W1(C1):** Existing tensor-based methods have primarily been applied to multi-view clustering tasks for mining higher-order feature correlations, while some matrix-based methods employ low-rank constraints to capture label semantic relevance. To the best of our knowledge, our proposed TMvML represents the first attempt to utilize tensor structures for MVML tasks, designed to simultaneously model both multi-view high-order correlations and multi-label co-occurrence patterns. The tensor formulation provides a natural and effective framework for capturing the intrinsic multi-dimensional relationships in MVML data that conventional matrix-based approaches cannot fully characterize.
>
> Although direct comparisons with other tensor methods are unavailable, we validate the effectiveness of our proposed Laplace Tensor Rank (LTR) by comparing TMvML with its variant TMvML-TNN, where we replace LTR with the traditional Tensor Nuclear Norm (TNN) for low-rank tensor structure approximation. Experimental results demonstrate that TMvML consistently outperforms TMvML-TNN across all datasets. This compelling evidence proves that our proposed LTR is more effective than traditional TNN in modeling the complex high-order correlations in multi-view multi-label learning tasks.
>
> |  |    | Emotions | Yeast | Corel5k | Plant| Espgame |Human|
> |-|-|-|-|-|-|-|-|
> |TMvML|AP| **0.811±0.020** | **0.771±0.008** | **0.440±0.008** | **0.608±0.007** | **0.306±0.001** | **0.631±0.010**|
> |TMvML-TNN|AP|0.738±0.014|0.747±0.014|0.382±0.015|0.601±0.017|0.270±0.020|0.621±0.007|
> |TMvML|Cov| **0.300±0.070** | **0.460±0.002** | **0.266±0.006** | **0.169±0.013** | **0.409±0.008** | **0.150±0.003** |
> |TMvML-TNN|Cov|0.344±0.032|0.467±0.009|0.279±0.003|0.171±0.012|0.452±0.009|0.162±0.005|
>
>
>
> **W2(C2):**  The introduction of the additional exponential term e^δ in the modified Laplace function, $f_{\mathrm{LTR}}(x)=1-\exp\left(-\frac{e^\delta x}{\delta}\right)$, provides several key advantages over the original Laplace function, $f_{\mathrm{Laplace}}(x)=1-\exp\left(-\frac{x}{\delta}\right)$.
>
> - The modified function offers enhanced flexibility by allowing dynamic adjustment of the growth rate and magnitude through $e^δ$. When $δ$ is large, $e^δ$ amplifies $x$, making the function grow faster for small singular values. When δ is small, the effect of $e^δ$ is reduced, and the function behaves similarly to the original Laplace function. This adaptability makes the modified function more versatile in handling different data distributions.
>
> - The modified function exhibits faster convergence for large values of $x$ due to the exponential scaling $e^δ$, improving optimization efficiency in tasks involving large-scale data or high-dimensional tensors.
>
> Thank you again for this valuable feedback. Please let us know if there is any additional information we can provide to assist with your evaluation.
>
> **W3(Q1):** We agree that theoretical guarantees for convergence are critical for ensuring the reliability and robustness of the optimization framework. We have added formal theoretical proofs ensuring convergence to a stationary point and reported the convergence theorem and its detailed proof in our rebuttal to **Reviewer miAu**.

---

### Official Review · Reviewer_miAu · 2025-03-10

**Overall Recommendation:** 3

**Summary:**

This paper proposes a Tensorized Multi-View Multi-Label Classification (TMvML) method to address the limitations of existing approaches that independently model cross-view consistent correlations and multi-label semantic relationships in MVML learning. The method reconstructs multi-view multi-label mapping matrices into a tensor classifier, where tensor rotation and low-rank constraints are jointly applied to unify view-level feature consistency and label-level semantic co-occurrence. Moreover, Laplace Tensor Rank is designed as a tight surrogate of tensor rank to capture high-order fiber correlations. The experimental results demonstrate the effectiveness of the proposed framework.

**Claims And Evidence:**

In this paper, the claims are supported:

1. TMvML’s superiority over SOTA is validated via experiments (Table 2) and statistical tests (Friedman/Bonferroni-Dunn).

2. LTR’s effectiveness is justified theoretically and empirically (Figure 3, Figure 5).

**Essential References Not Discussed:**

There are no essential references missing or overlooked in the paper's discussion of related work.

**Experimental Designs Or Analyses:**

1. This paper conducts extensive experiments on several datasets, and the experimental results demonstrate the effectiveness of the proposed framework.

2. Ablation studies and parameter sensitivity tests rigorously validate the contributions of LTR and hyperparameter stability.

**Methods And Evaluation Criteria:**

This paper uses the tensorized classifier for MVML, as tensors naturally model multi-dimensional relationships. The rotation operation cleverly reorients the tensor to align label-view interactions, addressing a key limitation of matrix-based methods. This paper evaluates the proposed framework on widely-used MVML benchmark datasets with five standard evaluation metrics and the results demonstrate the effectiveness of the method.

**Other Comments Or Suggestions:**

Please see the points under Weaknesses above.

**Other Strengths And Weaknesses:**

Strengths:

- Originality: This paper applies the tensor framework to the MVML problem for the first time, advancing the field by unifying cross-view consistency and label semantics in a single tensor classifier. This addresses a critical gap in existing MVML methods, which often handle these aspects independently.

- Experiments: The experiments are sufficient and the effectiveness of the proposed method is substantiated through these experiments.

- Clarity: This paper is well organized and the proposed method is clearly written to understand. All experiments details are provided and the codes are also released.

Weaknesses：

- In Section 5.3, the authors provide a textual analysis of the convergence behavior of TMvML, supported by empirical convergence curves (Figure 7). While the empirical results demonstrate stable convergence across datasets, the paper would benefit from theoretical guarantees to further strengthen the credibility of the optimization process.

**Questions For Authors:**

Please see the points under Weaknesses above.

**Relation To Broader Scientific Literature:**

The paper builds on foundational works in tensor-based multi-view learning and multi-label classification, but it uniquely integrates these two paradigms through a unified tensorized framework. It applies the tensor framework to the MVML problem for the first time.

**Theoretical Claims:**

Theorem 3.1 is correct and clearly proven.

---

> ### Author Rebuttal · Authors · 2025-03-29
>
> **W1**: The convergence of TMvML is guaranteed through the validation presented in Theorem 1, with comprehensive and rigorous proof below.
>
> **Theorem 1**: Let $\\{\mathcal{P}_k = ({\bf Z}_k^{v}, {\bf E}_k^{v}, {\bf A}_k^{v}, {\bf W}_k^{v}, {\bf B}_k^{v}, \mathcal{C}\_k, \mathcal{G}\_k)\\}\_{k=0}^{\infty}$ be the sequence generated by Algorithm 1, then the sequence $\{\mathcal{P}_k\}$ meets the following two principles:
>
> 1). $\\{\mathcal{P}_k\\}$ is bounded;
>
> 2). Any accumulation point of $\\{\mathcal{P}_k\\}$ is a KKT point of Algorithm 1.
>
> To prove Theorem 1, we first introduce two lemmas.
>
> **Lemma 1** [1]: Let $\mathcal{H}$ be a real Hilbert space with inner product $\langle \cdot, \cdot \rangle$, norm $\\|\cdot\\|$, and dual norm $\\|\cdot\\|^{dual}$. For $y \in \partial \\|x\\|$, we have $\\|y\\|^{dual} = 1$ if $x \neq 0$ and $\\|y\\|^{dual} \leq 1$ if $x = 0$.
>
> **Lemma 2** [2]: Let $F: \mathbb{R}^{m \times n} \to \mathbb{R}$ be defined as $F(\mathbf{X}) = f(\sigma(\mathbf{X}))$, where ${\bf X} = {\bf U} \mathrm{Diag}(\sigma({\bf X})) {\bf V}^T$ is the SVD of ${\bf X}$, $r = \min(m,n)$, and $f: \mathbb{R}^r\to\mathbb{R}$ is differentiable and absolutely symmetric at $\sigma(\mathbf{X})$.  Then,
>
> $\frac{\partial F(\mathbf{X})}{\partial\mathbf{X}}=\mathbf{U}\mathrm{Diag}(\partial f(\sigma(\mathbf{X}))) \mathbf{V}^T.$
>
> where $\partial f(\sigma(\mathbf{X})) = (\frac{\partial f(\sigma_1(x))}{\partial \mathbf{X}}, \dots, \frac{\partial f(\sigma_r(x))}{\partial\mathbf{X}}).$
>
> **Proof of the first part**: On the $k+1$ iteration, from the updating rule of $ \mathbf{E}_{k+1}$, the first-order optimal condition should be satisfied.
>
> $0=\alpha\partial\left\\|{\bf E}_{k+1}^v\right\\|\_{2,1}+\mu_k({\bf E}\_{k+1}^v-({\bf X}^v-{\bf{Z}\_{k+1}}^v{\bf A}^v+{\bf B}\_k^v/\mu\_k))=\alpha\partial\left\\|{\bf E}\_{k+1}^v\right\\|\_{2,1}-{\bf B}\_{k+1}^v,$
>
> Thus, we have
>
> $\frac{1}{\alpha}[\mathbf{B}\_{k+1}^{v}]\_{i,j}=\partial\left\\|\left[\mathbf{E}\_{k+1}^{v}\right]\_{:,j}\right\\|\_{2},$
>
> The $\ell\_{2}$ norm is self-dual, so based on the Lemma 1, we have $\left\\|\frac{1}{\alpha}[\mathbf{B}\_{k+1}^{v}]\_{:,j}\right\\|\_{2}\geq1$. So the sequence $\\{\mathbf{B}\_{k+1}^{v}\\}$ is bounded.
>
> Next, according to the updating rule of $\mathcal{G}$, the first-order optimality condition holds:
>
> $\partial \\|\mathcal{G}\_{k+1}\\|\_{LTR}=\mathcal{C}\_{k+1}$
>
> Let $\mathcal{U} * \mathcal{S} * \mathcal{V}^{T}$ be the t-SVD of tensor $\mathcal{G}$. Based on Lemma 2 and the definition of LTR, we have:
>
> $\\|\partial\\|\mathcal{G}\_{k+1}\\|\_{\text{LTR}}\\|\_F^2\leq\frac{e^{2\delta}\min(n\_1,n\_2)}{\delta^2 n\_3^2}$
>
> Thus, $\\{\mathcal{C}\_{k+1}\\}$ is bounded.
>
> Based on the iterative method constructed in the algorithm, we can deduce:
>
> $\mathcal{L}\_{k}(\mathbf{Z}\_{k+1}^v, \mathbf{E}\_{k+1}^v, \mathbf{A}\_{k+1}^v, \mathbf{W}\_{k+1}^v, \mathcal{G}\_{k+1}, \mathbf{B}\_k^v, \mathcal{C}\_k) \leq \mathcal{L}\_{k-1} + \frac{\rho\_k + \rho\_{k-1}}{2 \rho\_{k-1}^2} \\| \mathcal{C}\_k - \mathcal{C}\_{k-1} \\|\_F^2 + \frac{\mu\_k + \mu\_{k-1}}{2 \mu\_{k-1}^2} \sum\_v \\| \mathbf{B}\_k^v - \mathbf{B}\_{k-1}^v \\|\_F^2$
>
> Summing both sides results in: $\mathcal{L}\_k$ is bounded, and consequently, all its components are bounded, including  $\\| \mathcal{G}\_{k+1}\\|\_{\text{LTR}}$. Then, the boundedness of $\\{\mathcal{G}\_{k+1}, \mathbf{Z}\_{k+1}, \mathbf{A}\_{k+1}\\}$ is easy to prove. Therefore, the sequence $\\{\mathcal{P}\_k\\}$ is bounded.
>
> **Proof of the second part:** By the Weierstrass-Bolzano theorem [3], there exists at least one accumulation point of the sequence $\\{{\mathcal{P}\_k}\\}_{k=1}^{\infty}$, denoted as $\mathcal{P}\_*$. Then, we have:
>
> $\lim_{k\to\infty}({\bf Z}\_k^v, {\bf E}\_k^v, {\bf A}\_k^v, {\bf B}\_k^v, {\bf W}\_k^v, \mathcal{C}\_k, \mathcal{G}\_k)=({\bf Z}\_*^v, {\bf E}\_*^v,  {\bf A}\_*^v, {\bf B}\_*^v, {\bf W}\_*^v, \mathcal{C}\_\*,\mathcal{G}\_\*)$
>
> From the update rules of $\mathbf{B}\_k^v$, $\mathcal{C}\_k$, with $\\{\mathbf{B}\_k^v\\}$, $\\{\mathcal{C}\_k\\}$ bounded and the fact $\lim_{k\to\infty}\mu_{k}=\infty$, we obtain:
>
> $\lim_{k\to\infty}(\mathbf{X}^v-\mathbf{Z}\_{k+1}^v \mathbf{A}\_{k+1}^v - \mathbf{E}\_{k+1}^v) = 0 \Rightarrow \mathbf{X}^v = \mathbf{Z}\_*^v \mathbf{A}\_*^v + \mathbf{E}\_*^v$
>
> $\lim_{k \to \infty} (\mathcal{W}\_{k+1} - \mathcal{G}\_{k+1}) = 0 \Rightarrow \mathcal{W}\_* = \mathcal{G}\_*$
>
> Combining the first-order optimality conditions of $\mathbf{E}\_{k+1}^v$ and $\mathcal{G}\_{k+1}$, we take the limit and obtain:
>
> $\mathbf{B}\_*^v=\alpha \partial\\|\mathbf{E}\_*^v\\|\_{2,1}, \quad \mathcal{C}\_\*=\beta\partial\\|\mathcal{G}\_\*\\|\_{LTR}$
>
> Therefore, the accumulation point $\mathcal{P}\_*$ generated by TMvML satisfies the KKT conditions.
>
> [1] The augmented lagrange multiplier method for exact recovery of corrupted low-rank matrices. arXiv, 2010.
>
> [2] Nonsmooth analysis of singular values. Part I: Theory. Set-Valued Analysis, 2005.
>
> [3] Introduction to real analysis, Wiley New York, 2000.

---

### Decision · Program_Chairs · 2025-05-01

**Decision:**

Accept (poster)

**Comment:**

This paper proposes Tensorized Multi-View Multi-Label Classification (TMvML), introducing a novel Laplace Tensor Rank (LTR) to jointly model high-order feature correlations and multi-label dependencies. The method is evaluated on six benchmark datasets, demonstrating superior performance over existing approaches.

TR provides a tighter low-rank surrogate than traditional tensor norms (e.g., TNN). Extensive experiments (ablation studies, sensitivity tests, statistical significance) support claims. The paper makes a significant contribution to MVML by introducing a unified tensor framework with rigorous theory and empirical validation. All reviewers were satisfied with the rebuttal, and the revised manuscript (with minor clarifications) will strengthen the final version.